# Kidney Injury Molecule-1 Is Upregulated in Renal Lipotoxicity and Mediates Palmitate-Induced Tubular Cell Injury and Inflammatory Response

**DOI:** 10.3390/ijms20143406

**Published:** 2019-07-11

**Authors:** Xueying Zhao, Xiaoming Chen, Yuanyuan Zhang, Jasmine George, Alyssa Cobbs, Guoshen Wang, Lingyun Li, Nerimiah Emmett

**Affiliations:** Department of Physiology, Morehouse School of Medicine, Atlanta, GA 30310, USA

**Keywords:** lipotoxicity, metabolic syndrome, inflammation, nephropathy

## Abstract

Diabetic nephropathy is increasingly recognized as a major contributor to kidney failure in patients with obesity and type 2 diabetes. This study was designed to identify the molecular mediators of kidney injury associated with metabolic syndrome with or without hyperglycemia. We compared renal gene expression profiles in Zucker lean (ZL), Zucker obese (ZO), and Zucker diabetic (ZD) rats using cDNA microarray with quantitative verification of selected transcripts by real-time PCR. Compared to the 20-week-old ZL control (glucose: 110 ± 8 mg/dL), both prediabetic ZO (glucose: 157 ± 11 mg/dL) and diabetic ZD (glucose: 481 ± 37 mg/dL) rats displayed hyperlipidemia and kidney injury with a high degree of proteinuria. cDNA microarray identified 25 inflammation and injury-related transcriptomes whose expression levels were similarly increased in ZO and ZD kidneys. Among them, kidney injury molecule-1 (KIM-1) was found to be the most highly upregulated in both ZO and ZD kidneys. Immunofluorescence staining of kidney sections revealed a strong correlation between lipid overload and KIM-1 upregulation in proximal tubules of ZO and ZD rats. In cultured primary renal tubular epithelial cells (TECs), administration of saturated fatty acid palmitate resulted in an upregulation of KIM-1, osteopontin, and CD44, which was greatly attenuated by U0126, an inhibitor of extracellular signal-regulated kinase (ERK)1/2. Moreover, knockdown of KIM-1 by siRNA interference inhibited palmitate-induced cleaved caspase-3, osteopontin, and CD44 proteins in primary TECs. Our results indicate that KIM-1 expression is upregulated in renal lipotoxicity and may play an important role in fatty acid-induced inflammation and tubular cell damage in obesity and diabetic kidney disease.

## 1. Introduction

Obesity and type 2 diabetes affect one third of adults in the United States and are the leading causes of end-stage renal disease. Vascular abnormality and renal damage are commonly associated with metabolic syndrome, which is characterized by insulin resistance, obesity, dyslipidemia, hypertension, and proinflammatory and prothrombotic states, with or without glucose intolerance and hyperglycemia. In humans, obesity and diabetes-related nephropathy manifest as a clinical syndrome consisting of albuminuria, progressive decline in renal function, and an increased risk for cardiovascular disease [1]. Evidence also suggests that tubular injury and inflammatory processes are involved in diabetic nephropathy progression. 

Plasma concentrations of fatty acids (FAs) are increased in patients with metabolic syndrome, obesity, and type 2 diabetes [1,2,3]. Free FAs circulate primarily in association with albumin. Saturated FAs, such as palmitate, in non-adipose tissue can activate inflammatory signaling pathways, leading to lipotoxicity and cell dysfunction [4]. Normally, a minimum amount of albumin-bound FAs is filtered through glomeruli and reabsorbed into the proximal tubules [5,6,7]. In albuminuric/proteinuric kidney disease, the damaged glomerulus allows albumin-bound FAs to be filtered and gain access to the previously unexposed proximal tubule luminal surface, where aberrant FA reabsorption could then occur. Excess FAs overload the proximal tubular cells, aggravating urinary protein-related tubulointerstitial damage. However, the exact mediator(s) responsible for FA-induced tubulointerstitial inflammation and renal dysfunction remain largely unknown.

Kidney injury molecule-1 (KIM-1) or T-cell Ig and mucin domain (TIM-1, also known as HAVCR1)**,** a type 1 transmembrane protein, is expressed at negligible levels in normal kidney tissues, but is massively induced in damaged proximal tubules after acute and chronic kidney injury [8,9,10]. We have previously reported that tubular expression and urinary excretion of KIM-1 were significantly increased in diabetic animals [11,12,13]. Recent studies also show that increased plasma and urinary KIM-1 levels are associated with increased risk of developing early renal decline in diabetic patients [14,15,16]. KIM-1 was negatively correlated with estimated glomerular filtration rate (eGFR), independently of diabetes duration and albumin excretion rate [17]. However, few studies have focused on the functional role and underlying mechanism for KIM-1 in obesity and diabetic kidney disease. 

The Zucker obese (ZO) and the Zucker diabetic (ZD) rat strains have been well characterized as animal models of metabolic syndrome. The ZD rat was derived from the ZO strain by inbreeding for the hyperglycemia phenotype, which only occurred in a subpopulation of the Zucker rat. Therefore, the genetic backgrounds for these rats are similar. Compared to the euglycemic ZO rats, all ZD males become hyperglycemic by 10 weeks of age, and glucose remains elevated throughout their life span [13]. Free FAs, triglycerides, and cholesterol levels are significantly higher in the euglycemic ZO and hyperglycemic ZD rats compared with lean littermate controls. The objectives of this study were to (1) identify renal injury and inflammation-related genes commonly impacted in prediabetic ZO and diabetic ZD kidneys by gene microarray, (2) determine the differential effects of albumin-bound saturated and unsaturated FAs on tubular epithelial cells by evaluating KIM-1 and other biomarkers of renal injury and inflammation, and (3) examine whether KIM-1 deficiency would attenuate FA-induced tubular cell damage and inflammatory response.

## 2. Results

### 2.1. Metabolic Parameters 

The changes of body weight in Zucker lean (ZL), ZO, and ZD rats from 12–22 weeks of age are depicted in Figure 1A. Compared to the age-matched ZL control (290 ± 6 g), 12-week-old ZO (407 ± 15 g) and ZD (356 ± 19 g) animals were significantly heavier. The ZO rats continued to gain weight fast and averaged 662 ± 37 g at 22 weeks of age. In contrast, the ZD rats displayed attenuated weight gain during this period. Body weight was not different between ZD rats (433 ± 26 g) and lean littermates (424 ± 12 g) at 22 weeks. As expected, blood glucose was only slightly increased in the ZO rats, whereas blood glucose was greatly elevated in ZD rats (Figure 1B). Compared to the lean controls, plasma cholesterol (Figure 1C) and triglyceride (Figure 1D) levels were significantly increased in ZO and ZD rats with the highest values observed in ZO rats. 

### 2.2. Kidney Injury in ZO and ZD Rats 

In the present study, metabolic syndrome-associated renal damage and dysfunction were evaluated by measuring urinary protein excretion and tubulointerstitial fibrosis score. As shown in Figure 1E, urinary protein excretion was significantly increased in 22-week-old ZO rats (446 ± 65 mg/day) compared with the lean controls (163 ± 23 mg/day). Hyperglycemia further increased urinary protein to 629 ± 45 mg/day. Similarly, collagen deposition and renal fibrosis were increased in the ZO rats and worsened with hyperglycemia (Figure 1F–G). The tubulointerstitial fibrosis scores were significantly increased in both the ZO (1.69 ± 0.29, *p* < 0.01) and ZD (2.75 ± 0.26, *p* < 0.001) rats compared with their lean littermates (0.20 ± 0.09) (Figure 1F).

### 2.3. Genes Commonly Impacted in ZO and ZD Kidneys 

To assess genes differentially expressed in ZO and ZD kidneys of 22-week-old rats compared with their lean littermates, the Genespring software was used. The expression profile of each experimental group was determined in three animals per group. At least a two-fold change was used to determine the differentially-expressed genes in the ZO and ZD kidneys compared to the lean group. By this criterion, 1679 from a total of 41,090 rat transcripts were differentially expressed between ZL and ZO rats, whereas 1357 probes were altered between ZL and ZD rats. As shown in Figure 2, ZO and ZD rats showed similar changes in the expression of 590 genes, of which 282 genes were upregulated and 308 genes downregulated. The functional categories in these overlapped transcripts included genes involved in cell movement and infiltration, cell-to-cell signaling and interaction, cell proliferation and morphology, inflammation, lipid and carbohydrate metabolism, organ fibrosis, and necrosis. The upregulated inflammation and injury-related genes included *KIM-1*, osteopontin (*OPN*), *CD44*, *MCP-1/CCL2*, and fibronectin-1 (Table 1 and Table 2). Among them, *KIM-1* was found to be the most highly upregulated gene in the kidneys of ZO and ZD rats. *KIM-1* mRNA was increased more than 20-fold in ZO and ZD kidneys compared to normal controls. Real-time PCR analysis validated that these transcripts were similarly affected in ZO and ZD kidneys (Figure 3).

### 2.4. An Upregulation of KIM-1 Was Associated with Lipid Accumulation and CD44 Induction in Injured Proximal Tubular Cells of ZO and ZD Rats 

KIM-1 upregulation has been shown to correlate with damage to the proximal tubule in both acute and chronic kidney diseases [18,19,20,21,22]. We have previously reported an induction of KIM-1 in proximal tubules of diabetic ZD rats [11,13,23]. Here, we expanded our study to examine the relationship between KIM-1 expression and lipid accumulation in renal tubules of ZO and ZD kidneys. BODIPY 493/503 stain of kidney sections was performed to assess lipid level in normal and injured renal tubules. As depicted in Figure 4A, lipid-containing vesicles in normal tubules were rare. Significantly increased lipid vesicles were observed in tubular epithelial cells of ZO and ZD kidneys. Apical expression of KIM-1 was apparent in tubules with high levels of intracellular lipid vesicles in both ZO and ZD kidneys, indicating a strong correlation between lipid accumulation and KIM-1 upregulation in injured renal tubules. 

During inflammatory kidney diseases, renal CD44 expression, which is generally absent in normal conditions, is markedly enhanced, particularly in injured proximal tubular epithelial cells. Double immunostaining for KIM-1 and CD44 showed that expression of these two proteins in normal ZL kidneys was barely detectable (Figure 4B). However, CD44 de novo protein expression was abundant in the basolateral membrane of proximal tubules with strong KIM-1 staining in both ZO and ZD kidneys. Although KIM-1 was mainly present on the apical membrane, we also observed a co-localization of KIM-1 and CD44 on the proximal tubule basolateral membrane, especially in the dilated renal tubules lined with flattened tubular epithelial cells. 

### 2.5. Saturated FA Palmitate Increased KIM-1, OPN, and CD44 Expression via the ERK1/2 Signaling Pathway in Primary Rat TECs 

Aberrant reabsorption of albumin and conjugated FAs has been shown to be toxic to proximal tubule epithelial cells and may contribute to nephropathy progression [6,7,24,25]. To evaluate the effects of FAs on primary rat TECs directly, the cells were incubated with unsaturated oleate (300 µM) or saturated palmitate (300 µM) for 24 h. Western blot analysis revealed differential effects of palmitate versus oleate on tubular expression of KIM-1, cleaved caspase-3, OPN, and CD44. Palmitate treatment resulted in a significant increase in KIM-1 protein in TECs. In contrast, a slight, but significant decrease in KIM-1 was detected in oleate (OA)-treated cells (Figure 5A,B). Moreover, an upregulation of cleaved caspase-3, CD44, and OPN protein expression was observed in cells treated with palmitate, but not oleate (Figure 5A,C–F).

Since ERK1/2 has been shown to regulate mouse KIM-1 expression physiologically and following ischemic and septic renal injury [26], we next examined whether palmitate-induced KIM-1 is also dependent on ERK1/2 activation. As shown in Figure 5A,B, pretreatment with U0126, a specific ERK1/2 inhibitor, resulted in a significant reduction of KIM-1 in palmitate-stimulated cells. Palmitate-induced OPN and CD44 upregulation was also abolished by ERK1/2 inhibition (Figure 5A,C,D), whereas cleaved caspase-3 was further increased by U0126 treatment. Western blot analysis confirmed an activation of ERK1/2 in PA-treated TECs, which was greatly inhibited by U0126 (Figure 5G–I). As depicted in Figure 5J–M, a decrease in KIM-1 and an increase in cleaved caspase-3 were also observed in albumin control cells following ERK1/2 inhibition by U0126.

### 2.6. KIM-1 Is Involved in Palmitate-Induced Caspase-3 Activation and Upregulation of OPN and CD44 

Our in vivo results showed a close relationship among lipid overload, KIM-1, and CD44 upregulation in activated TECs. We then examined the function role of KIM-1 in tubular cell activation upon palmitate stimulation. As shown in Figure 6A–C, knockdown of KIM-1 with siRNAs suppressed the production of cleaved caspase-3 in primary TECs treated with palmitate. In addition, palmitate-induced OPN and CD44 expression was attenuated by siRNA-mediated KIM-1 silencing (Figure 6D–F). 

## 3. Discussion

There is growing evidence that tubular injury and inflammatory processes are involved in nephropathy progression in obesity and type 2 diabetes. In the current study, we found that the transcriptional levels of genes related o kidney injury and inflammation were similarly impacted in both prediabetic ZO and diabetic ZD kidneys. An upregulation of KIM-1 in lipid-overloaded proximal tubules further supports a pathogenic role of albumin-conjugated FAs in tubular injury and nephropathy progression. Using an in vitro cell culture model, we showed that KIM-1 silencing attenuated palmitate-induced tubular cell damage and upregulation of CD44 and OPN.

Our microarray data showed that 590 genes were similarly differentially expressed in ZO and ZD kidneys. Functional classification further revealed that these overlapped transcripts were enriched in genes that regulate cell movement and infiltration, cell-to-cell signaling and interaction, cell proliferation and morphology, inflammation, lipid and carbohydrate metabolism, organ fibrosis, and necrosis. Gene microarray identified 25 injury and inflammation-associated genes, which were similarly upregulated in ZO and ZD kidneys. We verified the mRNA levels of KIM-1, OPN, MCP-1/CCL2, and fibronectin-1 by real-time PCR and showed that KIM-1 was the most highly upregulated in diseased kidneys. These findings are in agreement with our previous reports showing that KIM-1, MCP-1, and OPN were upregulated in the kidneys of diabetic ZD rats [11,13,23].

In chronic glomerular diseases characterized by albuminuria/proteinuria, filtered FAs bound to albumin are reabsorbed by the downstream proximal tubule. Accumulation of FAs and FA metabolites has been shown to be cytotoxic to proximal tubule epithelial cells [5,6,7,27] and may contribute to tubular atrophy and nephropathy progression [5,6,7]. Although ZO rats did not develop hyperglycemia throughout the study period, kidney injury, as evidenced by a progressive increase in proteinuria, was comparable to the diabetic ZD rats. Therefore, we propose that accumulation of albumin and/or conjugated FAs leads to an upregulation of genes related to kidney injury and inflammation. A close correlation between lipid overload and KIM-1 induction is supported by the findings that intensive KIM-1 signaling was mainly detected in the tubular epithelial cells filled with lipid-containing vesicles. Therefore, we postulate that lipid accumulation may play an important role in tubular cell injury and dysfunction associated with metabolic syndrome with or without hyperglycemia.

Normally, free FAs dissociate from albumin at the plasma membrane and are taken up by saturable, basolateral membrane transporters in proximal tubules [28,29]. Under pathologic circumstances, the possibility of simultaneous apical and basolateral proximal tubule FA uptake combined with increased FA synthesis [30,31] and diminished β–oxidation [32] could lead to intracellular FA accumulation and tubular atrophy through a lipotoxicity mechanism [6,7,33]. In the present study, we did not address whether an increase in tubular lipids results from a combination of increased uptake, increased synthesis, or decreased catabolism of FAs. Understanding the biological mechanisms and the long-term consequences of lipid accumulation in proteinuric kidney disease will undoubtedly require further studies. Furthermore, it would be interesting to examine the transporters involved in the uptake of FAs by basolateral and apical membranes in proteinuric kidney disease.

We next performed an in vitro cell culture study to test the hypothesis that albumin-conjugated FAs stimulate KIM-1, CD44, and OPN expression in tubular epithelial cells. KIM-1 is a sensitive biomarker of tubular cell injury in acute and chronic kidney diseases [18,19,20,21]. In line with the previous findings [6,7,24,25], we confirmed that renal proximal tubular cells are sensitive to saturated FA palmitate, but not unsaturated oleate by evaluating cell apoptosis and KIM-1 protein expression. Compared to delipidated BSA control, palmitate-stimulated cells demonstrated a significant upregulation of KIM-1 along with an increase in active caspase-3. In contrast, a slight reduction of KIM-1 was shown in oleate-treated tubular cells. Similarly, an increase in CD44 and OPN proteins was observed in cells treated with palmitate, but not oleate. Thus, our results support that saturated FAs are more lipotoxic to proximal tubular cells by promoting cell apoptosis and the production of inflammatory mediators.

Recently, ERK1/2 has been identified as a regulator of KIM-1 expression. ERK1/2 inhibition prevented KIM-1 transcription in vitro following toxicant exposure and attenuated increases in KIM-1 mRNA and protein in ischemic-reperfusion-induced kidney injury [26]. Therefore, we next examined whether palmitate-induced KIM-1 expression was also dependent on the ERK signaling pathway. Inhibition of ERK1/2 activation completely abolished palmitate-induced KIM-1. In fact, KIM-1 protein level in palmitate-treated cells was even lower than that in the BSA control group in the presence of U0126, suggesting that ERK1/2 regulates rat KIM-1 expression physiologically and following palmitate stimulation. Moreover, activation of the ERK1/2 signaling pathway is also required for palmitate-induced OPN and CD44 in renal tubular epithelial cells. These results are in agreement with previous reports showing that ERK1/2 regulates mouse KIM expression physiologically and following ischemic and septic renal injury [26] and that activated ERK1/2 increases CD44 in glomerular parietal epithelial cells [34,35]. Although our in vitro findings support that palmitate may induce the production of inflammatory mediators through the ERK signaling pathway, further in vivo studies are required to examine the signaling pathways involved in lipid overload-related tubular cell injury and inflammatory gene expression in proteinuric kidney disease.

In the present study, while palmitate alone activated ERK1/2 and increased cleaved caspase-3, pretreatment with U0126 significantly enhanced palmitate-induced apoptosis. In general, activation of ERK1/2 is known to induce cellular growth and inhibit cell death. For example, inhibition of the ERK1/2 pathway has been shown to increase ochratoxin A-induced necrosis and apoptosis in proximal tubular cells (opossum kidney and normal rat kidney epithelial) [36]. Administration of 10 or 20 μM U0126 to untreated or ciglitazone- and troglitazone-treated HeLa cells resulted in a decrease in Bcl-2 expression accompanied by the collapse of mitochondrial membrane potential, which in turn significantly enhanced apoptotic cell death [37]. Thus, it is reasonable that after activation of ERK1/2 by palmitate treatment, the ERK1/2 inhibitor U0126 enhanced palmitate-induced tubular cell apoptosis. Thus, ERK1/2 activation may protect the cell against palmitate-induced cell death.

As an early biomarker of tubular injury, KIM-1 may have various biological functions. For example, KIM-1 has been shown to possess anti-inflammatory effects by downregulating innate immunity and inflammation in the early stage of kidney injury [38]. A recent report also indicated that KIM-1 could promote macrophage migration and phenotype changes via the MAPK signaling pathway in kidney disease [39]. In the current study, an attenuation of palmitate-induced caspase-3 activation following KIM-1 silencing supports a functional role of KIM-1 in renal lipotoxicity. Knockdown of KIM-1 prevents the upregulation of OPN and CD44 upon palmitate stimulation, suggesting that KIM-1 may act as a mediator linking tubular cell injury to interstitial inflammation and fibrosis in proteinuric kidney disease. A pathogenic role of KIM-1 in linking acute tubular damage to chronic kidney disease has also been supported by recent reports showing that persistent KIM-1 expression resulted in chronic kidney damage in mice and zebrafish [20,40]. KIM-1 overexpression was associated with an activation of the mammalian target of rapamycin (mTOR) pathway, and inhibition of this pathway with rapamycin protected against KIM-1-induced kidney tubular injury [40]. Interestingly, mTOR inhibition has been shown to reduce lipotoxic macrophage cell death [41]. Therefore, further studies will focus on the elucidation of signaling pathways responsible for KIM-1-mediated renal lipotoxicity. 

In summary, we identified 25 inflammation and injury-associated genes that were similarly upregulated in prediabetic ZO and diabetic ZD kidneys in association with proteinuria and kidney injury. KIM-1 expression was closely correlated with lipid accumulation and CD44 induction in injured proximal tubules of ZO and ZD rats. KIM-1 deficiency prevented palmitate-induced cell apoptosis and tubular expression of CD44 and OPN. Our results indicate that inhibition of lipid overload and KIM-1 induction may represent a viable therapeutic strategy for the prevention and treatment of proteinuric kidney diseases.

## 4. Materials and Methods

### 4.1. Experimental Animals

Male Zucker lean (ZL, *n* = 6), Zucker obese (ZO, *n* = 6), and Zucker diabetic (ZD, *n* = 6) rats were purchased from Charles River Laboratories (Wilmington, MA, USA) at 8 weeks of age. Blood glucose was monitored using the Accu-chek glucometer (Roche, Mannheim, Germany) by tail-vein blood sampling. Twenty four-hour urine was collected, and urinary protein excretion was determined with a standard Bradford assay (Bio-Rad Laboratories, Hercules, CA, USA). At 20–22 weeks of age, the rats were anesthetized with pentobarbital (50 mg/kg, ip). Blood was collected by cardiac puncture, and kidney tissues were harvested for gene and protein analysis. Rats were housed in an animal care facility at the Morehouse School of Medicine. All animal protocols were approved (approval number 16-12) on 31 March 2016 by theMorehouse School of Medicine Animal Care and Use Committee and were in accordance with the requirements stated in the National Institutes of Health Guide for the Care and Use Laboratory Animals.

### 4.2. RNA Isolation

Total RNA was prepared from isolated kidney cortex by using ultra-pure TRIzol reagent according to the manufacturer’s instructions (GIBCO-BRL, Grand Island, NY, USA). The integrity of the RNA samples was examined using the Agilent 2100 Bioanalyzer (G2938A). RNA samples, without evidence of degradation were used for microarray and real-time PCR analyses.

### 4.3. Gene Microarray

Gene expression studies were performed using the 60-mer Whole Rat Genome Oligo Microarray kit (p/n G2519F-14879). Complementary RNA (cRNA) was generated and fluorescently labeled from 1 μg of total RNA in each reaction using the One-Color Agilent Low RNA Input Fluorescent Linear Amplification Kit (p/n 5188–5339). Hybridization was performed following the Agilent oligonucleotide microarray hybridization user’s manual. The relative fluorescent intensity values in the Cy3 channel were then extracted from images using the Agilent Feature Extraction software 9.5.1 (p/n G2567AA) (Ball CA and Awad IA 2005). The gene expression data were analyzed using Spotfire software. An average of three replicate samples was used for each experimental condition. Using the fold change filter, genes were defined as differentially expressed if the fold change was equal to or greater than 2 compared with the lean littermate control. Comparisons of gene expression across conditional groups were performed using Venn diagram comparisons and various clustering algorithms (i.e., K-means and/or hierarchical). Gene Ontology groupings were utilized in conjunction with Spotfire to identify pathways and functional groups of genes. The ingenuity program (Ingenuity Systems, Mountain View, CA, USA) was also utilized to identify networks of interacting genes and other functional groups.

### 4.4. Quantitative Real-Time PCR

Reverse transcription was performed on equal amounts of total RNA by using random hexanucleotide primers to produce a cDNA library for each sample. Real-time PCR reactions were run on an iCycler iQ Real-Time PCR Detection System using Taqman Universal PCR Master Mix (Applied Biosystems, P/N 4304437). *KIM-1*, osteopontin (*OPN*)/secreted phosphoprotein-1, fibronectin-1, monocyte chemotactic protein-1 (*MCP-1/CCL2*), and β-actin gene-specific Taqman probe and primer sets were obtained from Applied Biosystems as Assays-on-Demand (AOD) gene expression products. The AOD identification numbers were Rn00597703_m1 for *KIM-1*, Rn00563571_m1 for *OPN*, Rn00569575_m1 for fibronectin-1, Rn00580555_m1 for *MCP-1*, and 4331182 for the rat β-actin endogenous control. Each sample was run in triplicate, and the comparative threshold cycle (C_t_) method was used to quantify fold increase (2^–∆∆Ct^) compared with controls.

### 4.5. Light Microscopy of Kidney Sections

Perfusion-fixed paraffin kidney sections (5 μm) were stained with periodic acid-Schiff and hematoxylin for histological evaluation. Another set of paraffin-embedded kidney sections was stained with Masson’s trichrome method to identify collagen fibers. Each lesion was given a grade using a 5-point grading scale: Grade 1 = minimal, Grade 2 = mild, Grade 3 = moderate, Grade 4 = marked, Grade 5 = very marked. Four to six kidneys were studied from each treatment group in a blinded fashion, and semi-quantitative evaluation for tubular injury was determined.

### 4.6. Immunostaining

To examine the expression and distribution of KIM-1 and CD44, 5-µm cryostat kidney sections were incubated with one or two primary antibodies overnight: goat anti-KIM-1 (1:100; R&D Systems, Minneapolis, MN, USA) and/or mouse anti-CD44 (1:100; R&D Systems). The secondary antibodies were Alexa Fluor 488-conjugated donkey anti mouse IgG (1:200) or Alexa Fluor 555-conjugated donkey anti-goat IgG (1:200) from Jackson ImmunoResearch Laboratories (West Grove, PA, USA). BODIPY 493/503 staining was also performed to evaluate FA accumulation in renal tubules. As a negative control, the sections were exposed to nonimmune IgG (in replacement of primary antibodies) with the same secondary antibodies, and no specific staining was observed. After nuclear staining with DAPI, the slides were mounted with ProLong gold antifade reagent (Thermo Fisher Scientific, Waltham, MA, USA). The sections were observed and imaged by a Leica confocal microscope (Wetzlar, Germany).

### 4.7. Culture of Primary Tubular Epithelial Cells and FA Treatment

Six- to ten-week-old Sprague–Dawley rat kidneys were dissected, and tubular fragments were collected and cultured in growth medium containing F12/DMEM (5 mM D-Glucose), 10% fetal bovine serum, penicillin, and streptomycin (Life Technologies, Carlsbad, CA, USA) until the tubular cells reached 80–90% confluency (Days 6–7). The cells were washed with serum-free DMEM and treated with saturated FA palmitate or unsaturated FA oleate (Sigma Aldrich Inc., St. Louis, MO, USA) complexed with 0.3% essential fatty acid-free bovine serum albumin (BSA, A7030, Sigma Aldrich Inc.) for 24 h. To evaluate the role of ERK1/2 MAP kinases, the cells were exposed to palmitic acid in the presence of U0126 (1,4-diamino-2,3-dicyano-1,4-bis (2-amino phenylthio) butadiene) (Sigma Aldrich Inc.), a selective ERK1/2 inhibitor. Cell lysates were prepared using RIPA lysis buffer with a cocktail of protease inhibitors (Sigma Aldrich Inc.).

### 4.8. RNA Interference

Primary cultured TECs were transfected with a validated non-targeting small interfering RNA (siRNA) control or with a siRNA specific for rat KIM-1 (Thermo Fisher Scientific). siRNAs delivery was performed using Lipofectamine RNAiMAX reagent (Thermo Fisher Scientific) according to the manufacturer’s instructions.

### 4.9. Western Blot Analysis

Cell lysates were separated by 10% SDS-PAGE and transferred electrophoretically to a nitrocellulose membrane. The blots were incubated with primary antibodies for KIM-1, cleaved caspase-3 (Cell Signaling Technology, Danvers, MA, USA), OPN (Thermo Fisher Scientific), CD44, phospho-ERK1/2 (Cell Signaling Technology), ERK1/2 (Cell Signaling Technology), or β–actin (Sigma Aldrich Inc.). The secondary antibodies were HRP-conjugated anti-goat IgG, anti-sheep IgG, or anti-rabbit IgG (ThermoFisher Scientific). Detection was accomplished using enhanced chemiluminescence Western blotting (ECL, GE Healthcare, Piscataway, NJ, USA). Relative band intensity was measured densitometrically by ImageJ software with β-actin as an internal control.

### 4.10. Statistical Analysis

Data are expressed as the mean ± SEM. Student’s *t*-test was used for comparison between the two groups. Comparisons among multiple groups were performed by one-way ANOVA and the Newman–Keuls post hoc test. Statistical significance was set at *p* < 0.05.

## Figures and Tables

**Figure 1 ijms-20-03406-f001:**
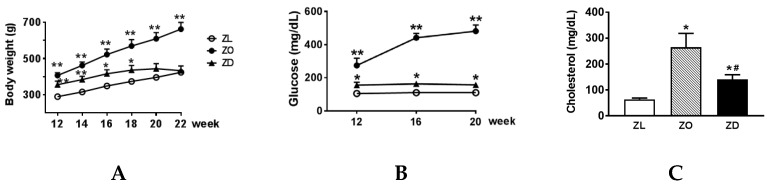
Metabolic abnormalities and increased kidney injury in Zucker obese (ZO) and Zucker diabetic (ZD) rats. Body weight (**A**), blood glucose (**B**), cholesterol (**C**) and triglyceride (**D**), urinary protein (**E**), tubulointerstitial fibrosis score (**F**), and representative images of Masson’s trichrome staining (**G**) in normal Zucker lean (ZL), prediabetic ZO, and diabetic ZD rats. Original magnification, ×400. Values are the mean ± SEM. *n* = 6 animals/group. **p* < 0.05, ***p* < 0.01 vs. the ZL group, ^#^*p* < 0.05 vs. the ZO group.

**Figure 2 ijms-20-03406-f002:**
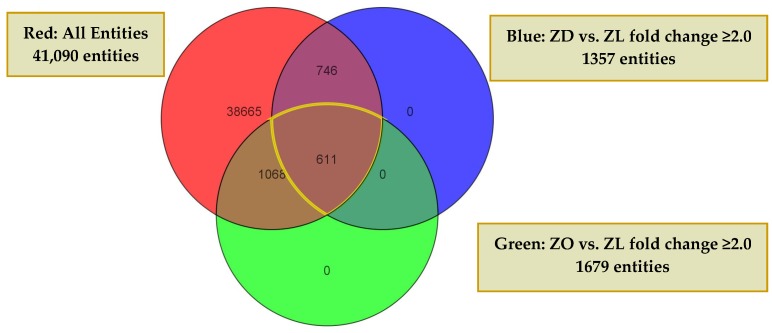
The genes were significantly altered in kidney cortex of Zucker rats. Circle graphs show the genes significantly altered (≥2 fold versus ZL) in kidney cortex of ZO (green) and ZD (blue) rats compared to ZL controls. Among them, 611 genes were commonly altered in the kidneys of ZO and ZD rats. *n* = 3 animals per group.

**Figure 3 ijms-20-03406-f003:**
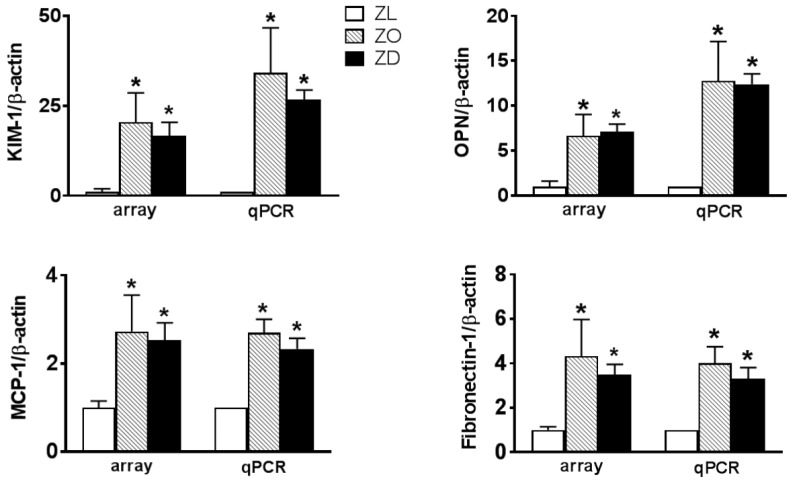
Increased KIM-1, OPN, MCP-1, and fibronectin-1 mRNA levels in ZO and ZD kidneys. mRNA levels of KIM-1, OPN, MCP-1, and fibronectin-1 in the kidney cortex of ZL, ZO, and ZD rats were measured by gene microarray (array) and quantitative real-time PCR (qPCR). mRNA fold changes were calculated using β–actin as an internal control. Values are the mean ± SEM. *n* = 3–6 animals/group. **p* < 0.05 vs. the ZL group.

**Figure 4 ijms-20-03406-f004:**
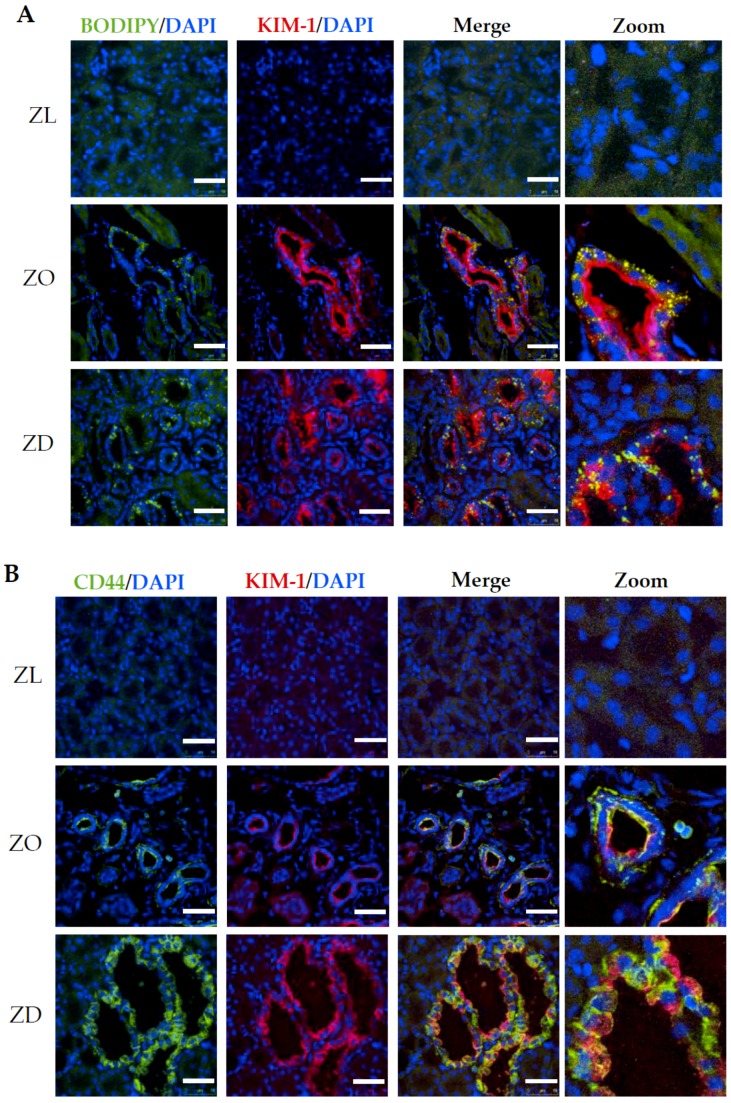
Immunofluorescence staining of lipid, KIM-1, and CD44 in Zucker rat kidneys. (**A**) Representative images show increased KIM-1 (red) expression in proximal tubular cells overloaded with intracellular lipids (BODIPY493/503, green) in kidney sections of both ZO and ZD rats compared to ZL controls. (**B**) De novo expression of CD44 (green) was observed in KIM-1-positive proximal tubules in ZO and ZD kidneys. DAPI was used for nuclear staining (blue). Bar: 50 µm.

**Figure 5 ijms-20-03406-f005:**
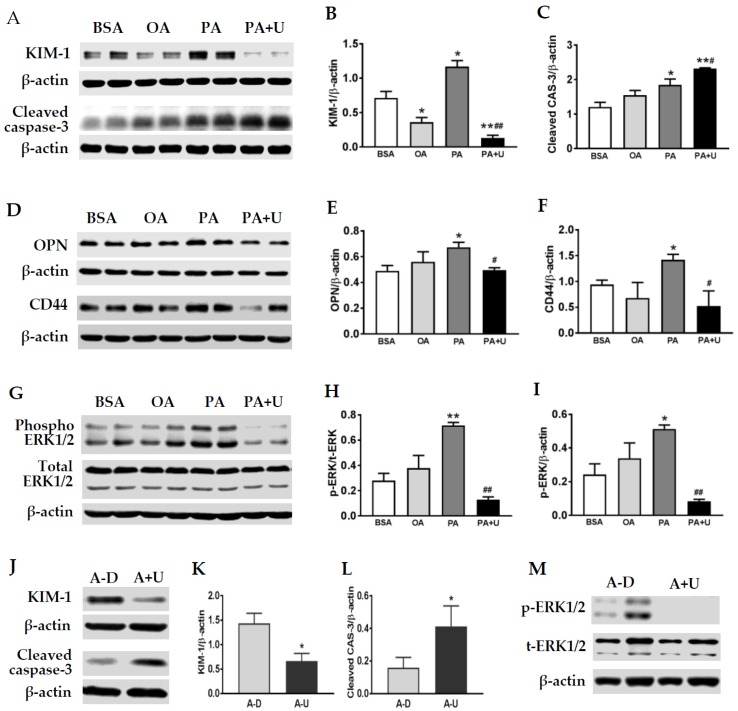
Increased KIM-1, cleaved caspase-3, OPN, and CD44 proteins in palmitate-treated TECs. Compared to the albumin (BSA) control, representative Western blot images and quantitative bar graphs show that KIM-1 (**A**,**B**), cleaved caspase (CAS)-3 (**A**,**C**), OPN (**D**–**E**), and CD44 (**D**,**F**) proteins were significantly increased in primary rat tubular epithelial cells (TECs) treated with palmitate (PA) but not oleate (OA). The phospho-ERK (p-ERK)1/2 level was significantly higher in TECs treated with PA for 24 h (**G**–**I**). Pretreatment of TECs with U0126 (U) inhibited ERK1/2 phosphorylation and abolished palmitate-induced KIM-1, OPN, and CD44 upregulation. In albumin control cells, U0126 (A-U) also reduced KIM-1 (**J**,**K**) and increased cleaved CAS-3 (**J**,**L**) compared to the DMSO vehicle group (**A**–**D**), which was associated with an inhibition of ERK1/2 activation (**M**). Values are the mean ± SEM. *n* = 3–4 individual experiments/group. **p* < 0.05, ***p* < 0.01 vs. the BSA control; ^#^*p* < 0.05, ^##^*p* < 0.01 vs. PA treatment.

**Figure 6 ijms-20-03406-f006:**
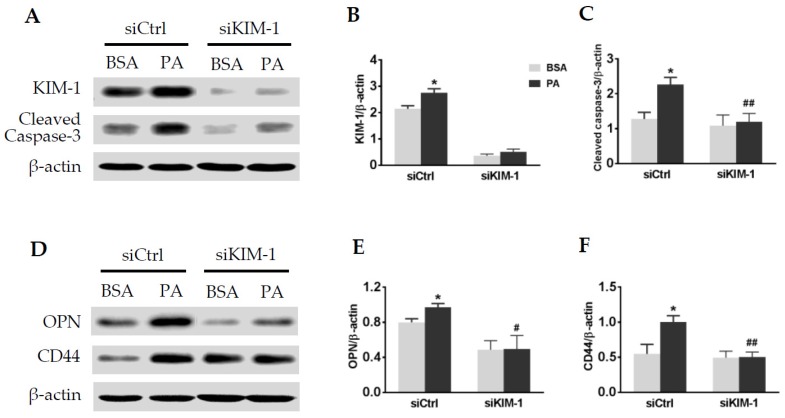
Knockdown of KIM-1 inhibited palmitate-induced cleaved caspase-3, OPN, and CD44 proteins in primary TECs. Knockdown of KIM-1 (**A**,**B**) by siRNA interference prevented an elevation of cleaved caspase-3 (**A**,**C**), OPN (**D**,**E**), and CD44 (**D**,**F**) upon palmitate stimulation. Values are the mean ± SEM. *n* = 3–4 individual experiments/group. **p* < 0.05 vs. the BSA control; ^#^*p* < 0.05, ^##^*p* < 0.01 vs. the siRNA control (siCtrl).

**Table 1 ijms-20-03406-t001:** Genes that play roles in inflammation and cell proliferation whose expression in the kidney was similarly altered in Zucker obese (ZO) and Zucker diabetic (ZD) rats.

Gene	Fold-Change		GenBank ID
	ZO	ZD	
Interleukin 24	32.02	19.73	NM_133311
Alpha-2-macroglobulin	11.92	4.60	NM_012488
Adrenergic receptor, alpha 1d	10.61	7.60	NM_024483
Early growth response 2	6.31	3.41	NM_053633
Kininogen 1	4.54	6.31	NM_012696
Activating transcription factor 3	3.68	2.16	NM_012912
Fibronectin 1	3.47	3.15	NM_019143
Poliovirus receptor	2.86	2.56	NM_017076
Chemokine (C-X-C motif) ligand 1	2.73	2.81	NM_030845
Chemokine (C-X-C motif) ligand 2	2.62	4.95	NM_053647
Myelocytomatosis viral oncogene homolog (avian)	2.61	2.29	NM_012603
Fos-like antigen 1	2.46	2.45	NM_012953
Nuclear protein 1	2.39	2.13	NM_053611
Lysyl oxidase	2.17	2.10	NM_017061
Parathyroid hormone-like peptide	2.16	2.04	NM_012636

**Table 2 ijms-20-03406-t002:** Genes that play roles in fibrosis and kidney injury whose expression in the kidney was similarly altered in Zucker obese (ZO) and Zucker diabetic (ZD) rats.

Gene	Fold-Change		GenBank ID
	ZO	ZD	
**Tubular Necrosis Related**			
Kidney injury molecule 1 (HAVCR1)	21.30	20.67	NM_173149
Secreted phosphoprotein 1 (SPP1, osteopontin)	6.34	7.82	NM_012881
**Fibrosis and Organ Damage Related**			
LIM/homeobox protein Lhx2 (Homeobox protein LH-2)	4.98	2.69	ENSRNOT00000014452
Matrix metallopeptidase 12 (Mmp12)	4.55	4.38	NM_053963
Troponin T2, cardiac (Tnnt2)	3.86	3.01	NM_012676
Activating transcription factor 3 (Atf3)	3.68	2.16	NM_012912
Serine (or cysteine) peptidase inhibitor, clade E, member 1 (Serpine1)	3.65	2.86	NM_012620
Adenosine A1 receptor	3.35	3.16	ENSRNOT00000004602
CD44 antigen (CD44)	3.10	2.58	NM_012924
Interleukin 1 receptor antagonist (Il1rn)	2.57	2.27	NM_022194

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
