# Peer review of "Kidney Injury Molecule-1 Is Upregulated in Renal Lipotoxicity and Mediates Palmitate-Induced Tubular Cell Injury and Inflammatory Response"

_ijms, 2019, doi:10.3390/ijms20143406_

Round 1
Reviewer 1 Report
Please see the attached annotated manuscript for my specific comments. My general comments are listed below.
1) Figure 1: The authors need to show statistical significance for body weight (Figure 1A) and blood glucose (Figure 1B) graphs. Also, the title needs to be changed to include metabolic abnormalities apart from the kidney injury measurements depicted in the figure.
The authors also need to consider including the photomicrographs of the histological sections of ZL, ZO, and ZD rat kidneys.
2) Table 2: Move the table 2 legend to next page to make it appear with the complete table.
3) Figures 3 and 6: Move the figure legends close to figurings to get rid of the extra space.
4) Figure 4: If available, use images of higher resolution in place of those shown in Figure 4.
5) All figures: The axis titles are not legible, and the images have a low resolution. The authors need to increase the font size of the text and use high-resolution graphs and images.
6) Albuminuria vs. Proteinuria in Abstract and Discussion: Since the authors did not specifically measure albumin levels in urine, it is inappropriate to refer your urinary protein measurements using Bradford assay as albuminuria. Please use proteinuria when referring to the urine protein measurements.
7) Discussion: Cleaved Caspase-3 levels An explanation of this observation and its significance to ERK1/2 signaling in the kidney was addressed in the discussion. What is the possible explanation for the induction of cleaved caspase-3 in U0126-treated cells? This conundrum needs to be elaborated in the discussion.
8) Some minor corrections and suggested change of word choices are indicated in the attached annotated manuscript. Please refer to the attachment for details.

Author Response
June 22, 2019
Manuscript ID: ijms-531719
Response to the reviewer #1:
We thank the reviewer for his/her positive and insightful comments on the manuscript. We have carefully considered each comment and have modified the manuscript accordingly. We are confident that the revised manuscript is significantly improved as a result of these suggestions. Our responses to the specific comments are indicated below.
1) Figure 1: The authors need to show statistical significance for body weight (Figure 1A) and blood glucose (1B) graphs. Also, the title needs to be changed to include metabolic abnormalities apart from the kidney injury measurements depicted in the figure. The authors also need to consider including the photomicrographs of the histological sections of ZL, ZO, and ZD rat kidneys.
As suggested by the reviewer, we have made the following changes:
a) We have added statistical significance for body weight (Figure 1A) and blood glucose (1B)
graphs.
b) We have modified and included both metabolic abnormalities and kidney injury
measurements in Figure 1 legend.
c) We have also included representative images of Masson’s trichrome staining of ZL, ZO, and ZD rat kidney sections (Figure 1G).
2) Table 2: Move the table 2 legend to next page to make it appear with the complete table.
Corrected and moved the table 2 legend to the same page with the complete table.
3) Figures 3 and 6: Move the figure legends close to figures to get rid of the extra space.
Corrected and moved the figure legends close to Figures 3 and 6.
4) Figure 4: If available, use images of higher resolution in place of those shown in Figure 4.
As suggested by the reviewer, we have used images of higher resolution in Figure 4.
5) All figures: the axis titles are not legible, and the images have a low resolution. The authors need to increase the font size of the text and use high-resolution graphs and images.
In the revised version of the manuscript, we have increased the font size of the text and used high-resolution graphs and images for all figures.
6) Albuminuria vs. Proteinuria in Abstract and Discussion: Since the authors did not specifically measure albumin levels in urine, it is appropriate to refer your urinary protein measurements using Bradford assay as albuminuria. Please use proteinuria when referring to the urine protein measurements.
This is a good point. We have used proteinuria when referring to the urine protein measurements in the revised manuscript.
7) Discussion: Cleaved Caspase-3 levels - An explanation of this observation and its significance to ERK1/2 signaling in the kidney was addressed in the discussion. What is the possible explanation for the induction of cleaved caspase-3 in U0126-treated cells? This conundrum needs to be elaborated in the discussion.
This is a very good point. In response to this comment, we have extended our discussion section to explain the induction of cleaved caspase-3 in U0126-treated cells and its significance to ERK1/2 signaling in the kidney (page 9-10, line 333-343). In addition, a 2-fold increase in cleaved caspase-3 was detected in BSA-treated cells in the presence of U0126. These results have been included in the Results section (page 7, line 240-241).
8) Some minor corrections and suggested change of word choices are indicated in the attached annotated manuscript. Please refer to the attachment for details.
We would like to thank the reviewer for those helpful suggestions. We have made all the corrections and changes as indicated in the attached annotated manuscript.
Reviewer 2 Report
General Comments:
Dr. Xueying Zhao and her colleagues studied renal injury and inflammation-related genes impacted in prediabetic ZO and diabetic ZD kidneys by gene microarray. They found that KIM-1 was increased in both ZO and ZD kidneys. In vitro, they discovered that saturated fatty acid palmitate resulted in an upregulation of KIM-1, osteopontin, and CD44, which was markedly attenuated by U0126, an inhibitor of the extracellular signal-regulated kinase (ERK)1/2. They conclude that KIM-1 expression is upregulated in renal lipotoxicity and may play an important role in fatty acid-induced inflammation and tubular cell damage in obesity and diabetic kidney disease. The experiments are well designed and executed. Most of the findings support the conclusion and also supported by their previous studies. However, there are some concerns that need to be addressed by the authors.
Specific Comments:
1. In Fig 1, the authors showed that the tubulointerstitial fibrosis score according to Masson’s trichrome staining, but they did not show the staining images. Please explain.
2. In Fig 4, The authors showed that KIM-1 was mainly present on the apical membrane, we also observed a co-localization of KIM-1 and CD44 on the proximal tubule basolateral membrane, especially in the dilated renal tubules lined with flattened tubular epithelial cells. Did you use a different marker of tubule cells to confirm it? KIM-1 may regulate the activation and formation of endothelial inflammasomes, which will contribute to inflammasome and injure.
3. In Fig 5, U0126 attenuated KIM-1 induced by PA, but significantly enhanced cle-Caspase-3. However, the author did not further explain or discuss. Base on the results, it is necessary to add more control group for the Western blot, example for BAS+U, OA+U.
4. How can you confirm that your primary culture cells are epithelial cells? There are many kinds of cells in tubular fragments.
5. Most of the experiments were repeated 3-5 times. It is necessary to repeat more than 5 times, especially for in vitro study.
6. Did you check the difference between male and female mice?
7. The authors were trying to demonstrate that KIM-1 regulate cell injury and inflammatory response, but they did not show any evidence on cell injury and inflammatory except CD44 in vitro. Caspase-3 or CD44 is not enough to demonstrate cell injury and inflammatory.
8. Resolution of Figures needs to be increased.
Author Response
June 22, 2019
Manuscript ID: ijms-531719
Response to the reviewer #2:
We thank the reviewer for his/her insightful and constructive comments on the manuscript. We have carefully considered each comment and have modified the manuscript accordingly. We are confident that the revised manuscript is significantly improved as a result of these suggestions. Our responses to the specific comments are indicated below.
General Comments:
Dr. Xueying Zhao and her colleagues studied renal injury and inflammation-related genes impacted in prediabetic ZO and diabetic ZD kidneys by gene microarray. They found that KIM-1 was increased in both ZO and ZD kidneys. In vitro, they discovered that saturated fatty acid palmitate resulted in an upregulation of KIM-1, osteopontin, and CD44, which was markedly attenuated by U0126, an inhibitor of the extracellular signal-regulated kinase (ERK)1/2. They conclude that KIM-1 expression is upregulated in renal lipotoxicity and may play an important role in fatty acid-induced inflammation and tubular cell damage in obesity and diabetic kidney disease. The experiments are well designed and executed. Most of the findings support the conclusion and also supported by their previous studies. However, there are some concerns that need to be addressed by the authors.
We would like to thank the reviewer for this positive evaluation.
Specific Comments:
1) In Fig 1, the authors showed that the tubulointerstitial fibrosis score according to Masson’s trichrome staining, but they did not show the staining images. Please explain.
In response to this comment, we have included representative images of Masson’s trichrome staining on ZL, ZO, and ZD rat kidney sections in Figure 1G.
2) In Fig 4, the authors showed that KIM-1 was mainly present on the apical membrane, we (they?) also observed a co-localization of KIM-1 and CD44 on the proximal tubule basolateral membrane, especially in the dilated renal tubules lined with flattened tubular epithelial cells. Did you use a different marker of tubule cells to confirm it? KIM-1 may regulate the activation and formation of endothelial inflammasomes, which will contribute to inflammasome and injure.
This is a good point. We have recently shown a co-localization of CD44 and megalin in dilated renal proximal tubules of ZD rats [1]. Megalin is an endocytic receptor localized in the kidney proximal tubules as well as in the glomeruli [2,3].
3) In Fig 5, U0126 attenuated KIM-1 induced by PA, but significantly enhanced cle-Caspase-3. However, the author did not further explain or discuss. Based on the results, it is necessary to add more control group for the Western blot, example for BSA+U, OA+U.
This is a very good point. In response to this comment, we have extended the Discussion section to explain the induction of cleaved caspase-3 in U0126-treated cells (page 9-10, line 333-343). As proposed by the reviewer, we have run additional Western blot showing that U0126 also increased cle-Caspase-3 protein by approximately 2-fold in BSA-treated cells. These results have been included in the Results section (page 7, line 240-241).
4) How can you confirm that your primary culture cells are epithelial cells? There are many kinds of cells in tubular fragments.
We agree with the reviewer that there are many kinds of cells in tubular fragments. As indicated in our previous report [4], after 6-7 days, tubular outgrowth became progressively larger and formed a monolayer with polygonal morphology and homogeneous appearance. Moreover, we confirmed that these cells have epithelial characteristics by showing albumin uptake and megalin expression in primary cultured cells [4].
5) Most of the experiments were repeated 3-5 times. It is necessary to repeat more than 5 times, especially for in vitro study.
We understand the reviewer’s concern. In our cell culture study, 2-3 animals were used every time, and every treatment was repeated for more than 3 times.
6) Did you check the difference between male and female mice?
We have not checked the difference between male and female rats or mice.
7) The authors were trying to demonstrate that KIM-1 regulate cell injury and inflammatory response, but they did not show any evidence on cell injury and inflammatory except CD44 in vitro. Caspase-3 or CD44 is not enough to demonstrate cell injury and inflammatory.
This is a good point. We have preliminary data showing that KIM-1 silencing also inhibited MCP-1 expression and secretion by PA-treated cells. However, we were not able to complete the study due to a discontinuation of the old MCP-1 antibody. We have tried a new replacement suggested by the company. Unfortunately, the new antibody did not work. Right now, we need new cell protein samples for Western blot analysis of different MCP-1 antibodies as well as for the evaluation of other injury and inflammatory markers. Since we are using primary cultured tubular cells, we cannot complete our new experiments within the next few months. Therefore, we have included additional reference support and proposed future studies we are considering in the revised Discussion section (page 10, line 352-359).
8) Resolution of Figures needs to be increased.
In the revised version of the manuscript, we have increased the resolution of Figures.
Reference List
1. Zhao X, Chen X, Chima A, Zhang Y, George J, Cobbs A, Emmett N (2018) Albumin induces CD44 expression in glomerular parietal epithelial cells by activating extracellular signal-regulated kinase 1/2 pathway. J Cell Physiol . 10.1002/jcp.27477 [doi].
2. Kerjaschki D, Farquhar MG (1982) The pathogenic antigen of Heymann nephritis is a membrane glycoprotein of the renal proximal tubule brush border. Proc Natl Acad Sci U S A 79: 5557-5561. 10.1073/pnas.79.18.5557 [doi].
3. Kerjaschki D, Farquhar MG (1983) Immunocytochemical localization of the Heymann nephritis antigen (GP330) in glomerular epithelial cells of normal Lewis rats. J Exp Med 157: 667-686. 10.1084/jem.157.2.667 [doi].
4. Chen X, Cobbs A, George J, Chima A, Tuyishime F, Zhao X (2017) Endocytosis of Albumin Induces Matrix Metalloproteinase-9 by Activating the ERK Signaling Pathway in Renal Tubule Epithelial Cells. Int J Mol Sci 18. ijms18081758 [pii];10.3390/ijms18081758 [doi].
Reviewer 3 Report
I read with interest the manuscript by Zhao and co-workers. The research is of interest for readers. The experimental design is sound with the results and the conclusions. In my opinion is worthy to publication with only a little suggestion:
the authors also indicate in the materials and methods (not only in the figures) the animal number used for each experimental group.
Author Response
June 22, 2019
Manuscript ID: ijms-531719
Response to the reviewer #3:
We thank the reviewer for his/her positive and insightful comments on the manuscript. We have carefully considered each comment and have modified the manuscript accordingly. We are confident that the revised manuscript is significantly improved as a result of these suggestions. Our responses to the specific comments are indicated below.
General Comments:
I read with interest the manuscript by Zhao and co-workers. The research is of interest for readers. The experimental design is sound with the results and the conclusions. In my opinion is worthy to publication with only a little suggestion: the authors also indicate in the materials and methods (not only in the figures) the animal number used for each experimental group.
We would like to thank the reviewer for this positive evaluation. As suggested by the reviewer, we have also indicated the animal number used for each experimental group in the Materials and Methods section (page 10, line 369).
Round 2
Reviewer 2 Report
The authors did a fine job in addressing concerns, however, a few critical concerns have not figured out.
1. As shown in Figure 5A-B, pretreatment with U0126, a specific ERK1/2 inhibitor, resulted in a significant reduction of KIM-1 in the absence or presence of palmitate. Palmitate-induced OPN and CD44 upregulation were also abolished by ERK1/2 inhibition (Figure 5A, C-D), whereas cleaved caspase-3 was further increased by U0126 treatment. A 2-fold 241 increase in cleaved caspase-3 was also observed in BSA-treated cells in the presence of U0126.
The results are not matching with the figures. I suggested to add two control groups of BSA+U0126 and OA+U0126, the author described them, but they were not shown in Figures.
2. The authors were trying to demonstrate that KIM-1 regulate cell injury and inflammatory response, but they did not show any evidence on cell injury and inflammatory except CD44 in vitro. Caspase-3 or CD44 is not enough to demonstrate cell injury and inflammatory.
This is a good point. We have preliminary data showing that KIM-1 silencing also inhibited MCP-1 expression and secretion by PA-treated cells. However, we were not able to complete the study due to a discontinuation of the old MCP-1 antibody. We have tried a new replacement suggested by the company. Unfortunately, the new antibody did not work. Right now, we need new cell protein samples for Western blot analysis of different MCP-1 antibodies as well as for the evaluation of other injury and inflammatory markers. Since we are using primary cultured tubular cells, we cannot complete our new experiments within the next few months. Therefore, we have included additional reference support and proposed future studies we are considering in the revised Discussion section (page 10, line 352-359).
The authors tried to provide more data to show some evidence of cell injury and inflammatory. But I cannot understand why they only focus on MCP-1, because there are a lot of methods to shown injure (?) and inflammatory (Such as IL-1beta, IL-18) by ELISA or IHC.
Author Response
Responses to reviewer #2:
We would like to thank the reviewer for your insightful and constructive comments. We have carefully considered each comment and have modified the manuscript accordingly. Our responses to the specific comments are indicated below. Modifications made to the manuscript are highlighted as red underlined text.
1. As shown in Figure 5A-B, pretreatment with U0126, a specific ERK1/2 inhibitor, resulted in a significant reduction of KIM-1 in the absence or presence of palmitate. Palmitate-induced OPN and CD44 upregulation were also abolished by ERK1/2 inhibition (Figure 5A, C-D), whereas cleaved caspase-3 was further increased by U0126 treatment. A 2-fold 241 increase in cleaved caspase-3 was also observed in BSA-treated cells in the presence of U0126. The results are not mating with the figures. I suggested to add two control groups of BSA+U0126 and OA+U0126, the author described them, but they were not shown in Figures.
In the revised version of the manuscript, we have added additional data indicating that a decrease in KIM-1 and an increase in cleaved caspase-3 were also detected in albumin control cells following ERK1/2 inhibition by U0126. These results have been included in Figure 5J-M (page 7) and in the results section (page 8, line 250-252).
2. The authors tried to provide more data to show some evidence of cell injury and inflammatory. But I cannot understand why they only focus on MCP-1, because there are a lot of methods to shown injure (?) and inflammatory (Such as IL-1beta, IL-18) by ELISA or IHC.
We’d like to thank the reviewer for your insightful suggestion. We’ve performed additional Western blot analysis using the conditioned media that we’ve previously collected. Unfortunately, no specific protein band was detected for IL-1beta. Because we observed multiple non-specific bands including a predominant albumin band during IL-1beta antibody incubation, we felt that the conditioned media collected from albumin-treated cells may be not suitable for ELISA analysis. We also cannot complete the IHC analysis in the next few months since we need to order new animals and perform new cell culture study.
Round 3
Reviewer 2 Report
The authors did a good job in addressing concerns, although they can not finish it because of technique limitation.